

# Acetylation of N-terminus and two internal amino acids is dispensable for degradation of a protein that aberrantly engages the endoplasmic reticulum translocon

Sarah M. Engle[1,2,*], Justin J. Crowder[1,3,*], Sheldon G. Watts[1,4], Christopher J. Indovina[1], Samuel Z. Coffey[1,5] and Eric M. Rubenstein[1]

[1] Department of Biology, Ball State University, Muncie, IN, United States of America
[2] Immunology—Translational Science, Eli Lilly and Company, Indianapolis, IN, United States of America
[3] Center for Medical Education, Indiana University School of Medicine, Muncie, IN, United States of America
[4] Marian University College of Osteopathic Medicine, Indianapolis, IN, United States of America
[5] Medpace Reference Laboratories, Cincinnati, OH, United States of America
[*] These authors contributed equally to this work.

## ABSTRACT

Conserved homologues of the Hrd1 ubiquitin ligase target for degradation proteins that persistently or aberrantly engage the endoplasmic reticulum translocon, including mammalian apolipoprotein B (apoB; the major protein component of low-density lipoproteins) and the artificial yeast protein *Deg1*-Sec62. A complete understanding of the molecular mechanism by which translocon-associated proteins are recognized and degraded may inform the development of therapeutic strategies for cholesterol-related pathologies. Both apoB and *Deg1*-Sec62 are extensively post-translationally modified. Mass spectrometry of a variant of *Deg1*-Sec62 revealed that the protein is acetylated at the N-terminal methionine and two internal lysine residues. N-terminal and internal acetylation regulates the degradation of a variety of unstable proteins. However, preventing N-terminal and internal acetylation had no detectable consequence for Hrd1-mediated proteolysis of *Deg1*-Sec62. Our data highlight the importance of empirically validating the role of post-translational modifications and sequence motifs on protein degradation, even when such elements have previously been demonstrated sufficient to destine other proteins for destruction.

## INTRODUCTION

Regulated protein degradation is crucial for cellular homeostasis. Misfolded, mislocalized, or otherwise aberrant proteins are targeted for destruction by the ubiquitin-proteasome system (UPS) (*Finley et al., 2012*). In addition, many proteins (e.g., cyclins and transcription factors) undergo regulated UPS-dependent degradation to meet cellular or organismal needs (*Hickey, 2016*; *Nakayama & Nakayama, 2006*; *Rubenstein & Hochstrasser, 2010*).

Corresponding author
Eric M. Rubenstein,
emrubenstein@bsu.edu

Enzymes called ubiquitin ligases (E3s) covalently transfer multiple copies of the small polypeptide ubiquitin to substrate proteins, marking them for destruction by the proteasome (*Kleiger & Mayor, 2014*). E3s, or their associated protein co-factors, recognize degrons (degradation signals) within their substrates. Degrons vary widely and include amino acid sequences, post-translational modifications (PTMs), and other structural features (*Ravid & Hochstrasser, 2008*).

Aberrant or physiologically unstable proteins at the endoplasmic reticulum (ER) are targeted for proteasomal destruction by one of multiple ER-Associated Degradation (ERAD) mechanisms (*Ruggiano, Foresti & Carvalho, 2014*). In the budding yeast *Saccharomyces cerevisiae*, multiple E3s (each with mammalian homologs) participate in ERAD (*Zattas & Hochstrasser, 2015*). The ER-resident E3 Hrd1 (HRD1 and gp78 in mammals) recognizes proteins with *l*uminal or intra*m*embrane degrons (ERAD-L or ERAD-M substrates, respectively) (*Carvalho, Goder & Rapoport, 2006*; *Gauss, Sommer & Jarosch, 2006*; *Sato et al., 2009*). The ER-resident E3 Doa10 (TEB4/MARCH-VI in mammals) ubiquitylates soluble and transmembrane proteins with *c*ytosolic degrons (ERAD-C substrates) and participates in the degradation of some ERAD-M substrates (*Habeck et al., 2015*; *Huyer et al., 2004*; *Metzger et al., 2008*; *Ravid, Kreft & Hochstrasser, 2006*). Non-ER-resident E3s also contribute to the proteolysis of ER-localized proteins. For example, the cytosolic E3 Ubr1 redundantly targets some Doa10 and Hrd1 substrates for degradation (*Stolz et al., 2013*). Recently, we and others found that translationally stalled, *r*ibosome-*a*ssociated ER-targeted proteins (ERAD-RA substrates) are marked for degradation by the cytosolic ribosome quality control E3 Rkr1/Ltn1 (Listerin in mammals) (*Arakawa et al., 2016*; *Crowder et al., 2015*; *Von der Malsburg, Saho & Hedge, 2015*).

Other proteins are targeted for Hrd1-dependent degradation after they have persistently or aberrantly engaged the ER *t*ranslocon in a mechanism termed ERAD-T (*Rubenstein et al., 2012*). For instance, the primary component of mammalian low-density lipoproteins (LDLs), apolipoprotein B (apoB), arrests during translocation. Translocation proceeds upon interaction of the luminally exposed N-terminal portion of apoB with its lipid binding partners, and an LDL particle is assembled. When lipid binding is impaired, the Hrd1 homologue gp78 recognizes and targets the translocon-associated, unassembled apoB for proteasomal degradation (*Fisher, Khanna & McLeod, 2011*; *Liang et al., 2003*; *Pariyarath et al., 2001*; *Rutledge et al., 2009*; *Yeung, Chen & Chan, 1996*). Consistent with conservation of translocon-associated quality control mediated by ERAD machinery, Hrd1 marks for destruction an apoB variant that persistently associates with the translocon when expressed in yeast cells (*Hrizo et al., 2007*; *Rubenstein et al., 2012*).

Fusion of *Deg1* (the first 67 amino acids of the transcriptional repressor MATα2) to the N-terminus of the otherwise stable, two-transmembrane yeast protein Sec62 triggers aberrant post-translational translocation of the cytosolic N-terminal Sec62 tail (*Rubenstein et al., 2012*; Fig. 1A). Translocon occupancy by the *Deg1*-Sec62 fusion protein is likely to be prolonged, as a disulfide linkage forms between cysteine residues in *Deg1*-Sec62 and the translocon channel interior (*Scott & Schekman, 2008*). Following aberrant translocon engagement, Hrd1 targets *Deg1*-Sec62 for proteasomal degradation. Mutation of either amino acid participating in this disulfide linkage substantially abrogates Hrd1-dependent

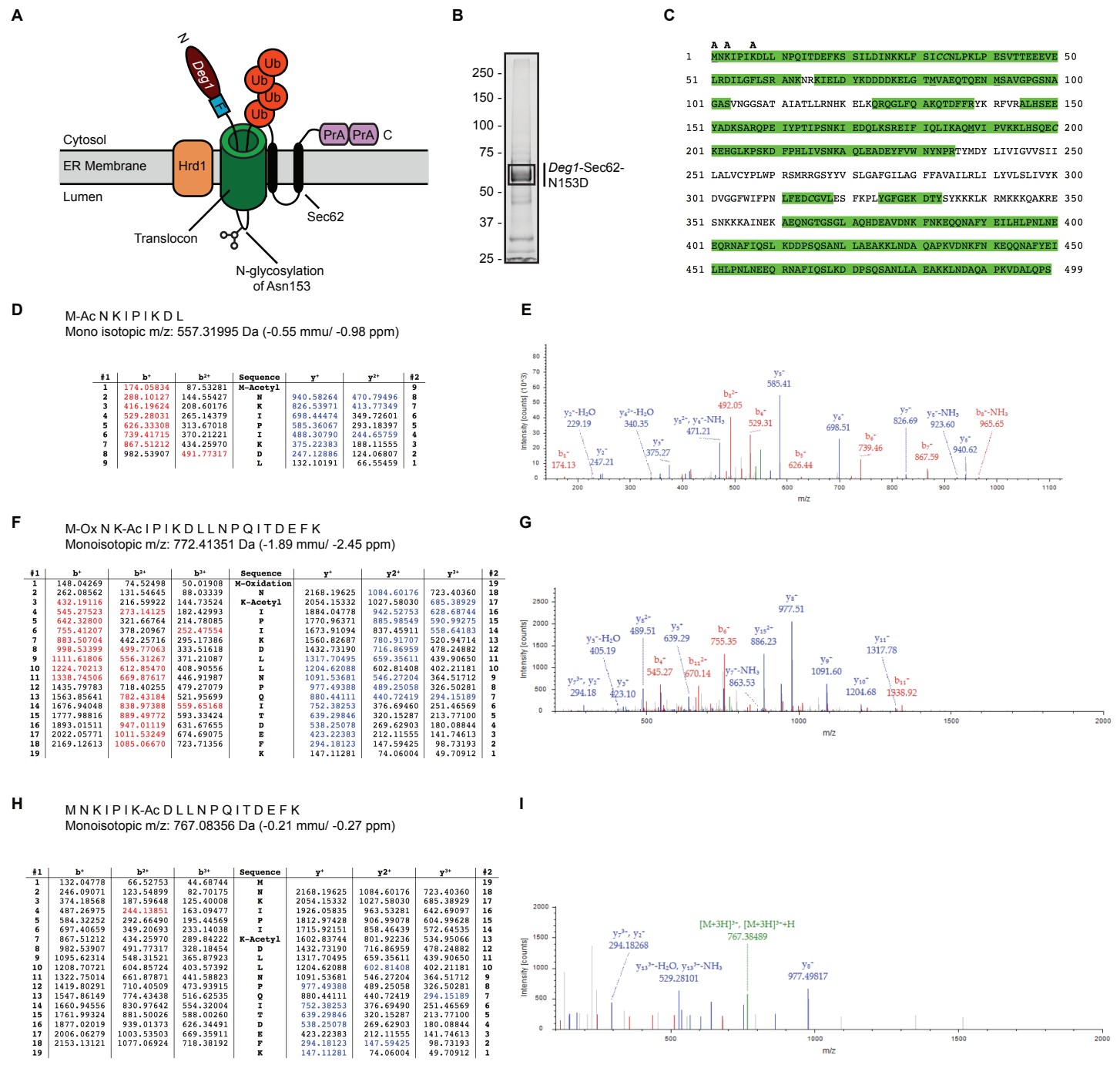

**Figure 1  Acetylation of *Deg1*-Sec62-N153D.**  (A) Schematic of *Deg1*-Sec62 following aberrant translocon engagement. *Deg1*-Sec62 possesses, in sequence, *Deg1* (the N-terminal 67 amino acids from the yeast transcriptional repressor MATα2), a Flag epitope (F), the two-transmembrane protein Sec62, and tandem copies of Protein A (PrA) from *Staphylococcus aureus*. Following co-translational insertion of the two transmembrane segments of Sec62 (black ovals), a portion of the cytosolic N-terminal tail aberrantly moves into the translocon via the post-translational translocation mechanism. Aberrant translocation is followed by N-linked glycosylation of Asn153 and Hrd1-dependent ubiquitylation (*Rubenstein et al., 2012*). 

**Figure 1 (…continued)**
(B) Purification of *Deg1*-Sec62-N153D. *Deg1*-Sec62-N153D was immunoprecipitated from lysates of *hrd1*Δ cells, separated by SDS-PAGE, and stained with GelCode Blue. Bands corresponding to *Deg1*-Sec62-N153D (demarcated by box) were excised, subjected to in-gel digestion, and evaluated by LC/ESI MS/MS. (C) Peptide map of *Deg1*-Sec62-N153D. High confidence peptide identifications from SEQUEST-HT (false discovery rate < 1%) were mapped to the *Deg1*-Sec62-N153D protein sequence. Regions highlighted in green are segments of the amino acid sequence identified by MS. Amino acids identified as acetylated are labeled "A". Underlined amino acids represent detection of methionine oxidation, a likely artifact of electrospray MS (*Chen & Cook, 2007*). Italicized amino acids represent cysteine residues modified by carbamidomethylation following iodoacetamide treatment. (D–I) Tandem mass spectra of proteolytically digested *Deg1*-Sec62-N153D were subjected to acetylpeptide identification using the database search algorithm SEQUEST-HT (part of Proteome Discoverer). The tables (D, F, H) display predicted fragment ions for the identified peptides and highlight in red and blue the b-and y-ions, respectively, identified in the fragment spectra (E, G, I). Red and blue peaks in the fragment spectra reflect tandem MS data that matched theoretical peptide fragment masses listed in the tables. Green peaks represent precursor ions or precursor ions with a neutral loss of water or ammonia. N, Amino terminus. C, Carboxyl terminus. Ub, ubiquitin. Ac, acetylation. Ox, oxidation.

degradation, suggesting that this persistent interaction is prerequisite to Hrd1-dependent degradation (*Rubenstein et al., 2012*). Thus, *Deg1*-Sec62 may function as a model substrate for proteins that persistently or aberrantly engage the ER translocon, such as apoB.

Degradation of many unstable proteins, including several Hrd1 substrates, depends on their PTM status (*Aebi et al., 2010*; *Clerc et al., 2009*; *Goder & Melero, 2011*; *Hirayama et al., 2008*; *Quan et al., 2008*; *Spear & Ng, 2005*; *Vashist et al., 2001*; *Xu et al., 2013*; *Zattas et al., 2013*). Following translocon engagement, *Deg1*-Sec62 becomes heavily modified in a manner that correlates with its Hrd1-dependent degradation. *Deg1*-Sec62 is decorated by N-linked glycosylation and O-linked mannosylation. However, preventing carbohydrate addition does not detectably affect Hrd1-dependent degradation or completely abolish PTM of *Deg1*-Sec62 (*Rubenstein et al., 2012*). These data suggest that the protein is modified in manners other than glycosylation that may be hypothesized to influence protein behavior and degradation.

In this study, we subjected *Deg1*-Sec62 to mass spectrometry (MS) analysis and found that it is acetylated on three amino acids (Met1, Lys3, and Lys7). Since N-terminal and internal acetylation regulates the degradation of other UPS substrates (*Hwang, Shemorry & Varshavsky, 2010*; *Lee et al., 2016*; *Nguyen et al., 2016*; *Zattas et al., 2013*), and behavior of the translocon-associated ERAD substrate apoB is influenced by acetylation (*Brown et al., 1980*; *Brown et al., 1979*; *Weisgraber, Innerarity & Mahley, 1978*), we hypothesized that Hrd1-dependent degradation of *Deg1*-Sec62 is influenced by acetylation. However, simultaneously inhibiting all three acetylation events did not affect Hrd1-mediated proteolysis of *Deg1*-Sec62. Thus, acetylation of Met1, Lys3, and Lys7 of the model unstable, translocon-associated *Deg1*-Sec62 protein is not required for targeting by the Hrd1 ubiquitin ligase.

## MATERIALS & METHODS

### Yeast and bacterial methods

Yeast strains and plasmids used in this study are presented in Tables 1 and 2, respectively. Standard methods were used for manipulation of yeast strains (*Guthrie & Fink, 2004*). Yeast cells were grown in synthetic defined medium (*Guthrie & Fink, 2004*). Galactose was included in growth medium instead of glucose to induce expression of *Deg1*-Sec62-N153D (driven by the *GAL1/10* promoter (*Johnston & Davis, 1984*)) for protein purification.

## PeerJ

**Table 1** Yeast strains used in this study.

| Name | Alias | Genotype | Source |
|------|-------|----------|--------|
| VJY20 | MHY6601 | MATα his3-Δ200 leu2-3,112 ura3-52 lys2-801 trp1-1 gal2 nat3Δ::TRP1 | Mark Hochstrasser and David Adle |
| VJY22 | MHY3032 | MATa his3Δ1 leu2Δ0 met15Δ0 ura3Δ0 hrd1Δ::kanMX4 | Tong et al. (2001) |
| VJY42 | MHY501 | MATα his3-Δ200 leu2-3,112 ura3-52 lys2-801 trp1-1 gal2 | Chen et al. (1993) |
| VJY172 | MHY6199 | MATα his3-Δ200 leu2-3,112 ura3-52 lys2-801 trp1-1 gal2 hrd1Δ::kanMX4 | Buchanan et al. (2016) |

**Table 2** Plasmids used in this study.

| Name | Alias | Description | Source |
|------|-------|-------------|--------|
| pVJ2 | pDN431 | CPY*-HA driven by PRC1 promoter<br>CPY* = CPY with G255R point mutation | Ng, Spear & Walter (2000) |
| pVJ27 | pRS316 | Empty vector | Sikorski & Hieter (1989) |
| pVJ312 | pRS316-GAL1/10-Deg1-Sec62-ProtA-N153D | Deg1-Sec62-ProtA-N153D driven by GAL1/10 promoter | Scott & Schekman (2008) |
| pVJ317 | pRS416-MET25-Deg1*-Sec62-ProtA | Deg1*-Sec62-ProtA driven by MET25 promoter<br>Deg1* = Deg1 with F18S, I22T point mutations | Rubenstein et al. (2012) |
| pVJ518 | pRS416-MET25-Deg1*-Sec62-ProtA-2R | Deg1*-Sec62-ProtA-2R driven by MET25 promoter<br>2R = K3R, K7R point mutations<br>Generated via site-directed mutagenesis of plasmid pVJ317 using primers VJR270 (5′ GAATAGAATTCCCATTAGAGACCTTTTAAATC 3′) and VJR271 (5′ GGTCTCTAATGGGAATTCTATTCATGGATCC 3′), which mutate Lys3 and Lys7 to Arg and introduce a silent EcoRI restriction site. | This study |
| pVJ527 | pRS416-MET25-Deg1*-Sec62-ProtA-3R | Deg1*-Sec62-ProtA-3R driven by MET25 promoter<br>3R = N2R, K3R, K7R point mutations<br>Generated via site-directed mutagenesis of plasmid pVJ518 using primers VJR280 (5′ CATGAGAAGAATACCCATTAGAGACCTTTTAAATC 3′) and VJR281 (5′ CTAATGGGTATTCTTCTCATGGATCCACTAG 3′), which mutate Asn2 to Arg and silently remove an EcoRI restriction site. | This study |

Notes.

All plasmids used in this study are yeast CEN plasmids, harbor the AmpR gene for selection of ampicillin-resistant E. coli, and the URA3 gene for selection for uracil prototrophy in yeast.

Plasmids were manipulated using PCR-based site-directed mutagenesis using primers presented in Table 2. Mutagenesis was confirmed by digestion with restriction enzymes and plasmid sequencing.

### Yeast cell harvest and immunoprecipitation

Overnight cultures of yeast cells grown in plasmid-selective medium containing glucose were diluted to an optical density at 600 nm ($OD_{600}$) of 0.2 in plasmid-selective medium containing galactose. Cultures were incubated, shaking, at 30 °C until the $OD_{600}$ reached 1.5. Approximately 15,000 $OD_{600}$ units (10 L) of logarithmically growing yeast cells were harvested by centrifugation in 1 L batches. Each batch of yeast cells was pelleted by centrifugation for 5 min at $3,800\times$ g, snap-frozen with liquid nitrogen, and stored at

−80 °C until lysis. Yeast cell pellets were thawed on ice and suspended in an equal volume of Triton X lysis buffer (500 mM NaCl, 50 mM HEPES pH 7.5, 5 mM EDTA, 1% Triton X-100) supplemented with protease inhibitors (1 tablet cOmplete Mini Protease Inhibitors; Roche Diagnostics, Indianapolis, IN, USA) per 5 mL buffer, 10 μg/mL pepstatin A) and phosphatase inhibitors (1 mM sodium fluoride, 0.5 mM sodium pyrophosphate). Volumes of glass beads approximately equal to the volumes of yeast cell pellets were added, and cell suspensions were vortexed 12 × 30 s at 4 °C, with two min on ice between rounds of vortexing. Lysed cells were centrifuged at 8,700× g for 10 min at 4 °C. The supernatants were subsequently centrifuged at 20,000× g for 10 min at 4 °C. The supernatants were preserved. Protein concentration was determined using the Bradford Assay.

Protein from cells expressing *Deg1*-Sec62-N153D (which possesses a FLAG affinity tag) was subjected to anti-FLAG immunoprecipitation. Briefly, 250 mg of protein extract was incubated, rotating, for 2.5 h at 4 °C in the presence of 600 μL of 50% αFLAG M2-conjugated agarose affinity gel (Sigma-Aldrich, St. Louis, MO) equilibrated with Triton X lysis buffer. The affinity gel was washed ten times with ice-cold immunoprecipitation wash buffer (500 mM NaCl, 50 mM HEPES pH 7.5, 5 mM EDTA, 1% Triton X-100, 0.1% sodium dodecyl sulfate). The *Deg1*-Sec62-N153D protein was eluted from the affinity gel by incubation with 75 mM 3xFLAG peptide (GenScript, Piscataway, NJ, USA) in a total volume of 2 mL of immunoprecipitation wash buffer, rocking, at 4 °C for 30 min. The affinity gel was pelleted by centrifugation at 6,900 × g, and the supernatant with eluted protein was transferred to fresh tubes and preserved. To reduce disulfide linkages and alkylate cysteine side chains prior to mass spectrometry, sequential addition of and incubation with dithiothreitol (5 mM final concentration, 56 °C, 30 min) and iodoacetamide (7 mM final concentration, room temperature, in a light-proof container) was performed.

Purified, reduced, and alkylated *Deg1*-Sec62-N153D protein was concentrated by trichloroacetic acid (TCA) precipitation. TCA was added to a final concentration of 20%. The sample was vortexed, incubated at 4 °C for 20 min, and centrifuged at 20,000 × g at 4 °C for 20 min. The pellet was washed once with 4 °C 10% TCA and once with ice-cold 100% high-performance liquid chromatography-grade acetone. The pellet was air-dried at room temperature for 10 min and fully dried in a vacuum concentrator for 30 min. The pellet was resuspended in 2X Laemmli Sample Buffer, heated at 95 °C for 5 min, and separated by sodium dodecyl sulfate-polyacrylamide gel electrophoresis (SDS-PAGE). The SDS-PAGE gel was stained in GelCode Blue (Thermo Scientific, Waltham, MA, USA), and bands corresponding to *Deg1*-Sec62-N153D were excised with a fresh razor blade.

## Mass spectrometry

Similar methods for mass spectrometry data acquisition and analysis have been described (*Hughes Hallett, Luo & Capaldi, 2015*). The gel slice containing *Deg1*-Sec62-N153D was washed 15 min each with water, 50% (v/v) acetonitrile, 100% (v/v) acetonitrile, 100 mM ammonium bicarbonate, and 100 mM ammonium bicarbonate in 50% (v/v) acetonitrile. The solution was removed, and the gel slice was dried by vacuum centrifugation. Protein in the dried gel slice was reduced by immersion in 10 mM dithiothreitol/100 mM ammonium bicarbonate at 56 °C for 45 min. The solution was removed and discarded. Protein in

the gel slice was alkylated by immersion in 55 mM iodoacetamide/100 mM ammonium bicarbonate in the dark at ambient temperature for 30 min. The solution was removed and discarded. The gel slice was washed with 100 mM ammonium bicarbonate for 10 min on a shaker. An equal amount of acetonitrile was added, and the slice was incubated, shaking, for 10 min. The solution was discarded and the gel slice was dried by vacuum centrifugation for 45 min. The gel slice was cooled on ice. The slice was immersed in an ice-cold solution of 12.5 ng/$\mu$L trypsin (Promega, Madison, WI) in 100 mM ammonium bicarbonate. After 45 min, the trypsin solution was discarded, and the gel slice was immersed in 50 mM ammonium bicarbonate for 16 h at 37 °C on a shaker. The sample was sedimented by microcentrifugation, and the supernatant was collected and retained. Peptides were extracted from the gel slice by immersing the slice in 0.1% (v/v) trifluoroacetic acid (TFA) and mixing at ambient temperature for 30 min. An equal volume of acetonitrile was added, and the sample was mixed for an additional 30 min. The sample was sedimented by microcentrifugation, and the supernatant from this extraction was pooled with the post-digestion supernatant. This pool was divided into 4 tubes and concentrated by vacuum centrifugation. One tube was further digested with chymotrypsin (Roche Diagnostics, Indianapolis, IN, USA) by resuspending the tryptically digested peptides in a solution of 100 mM Tris–HCl/10 mM calcium chloride (pH 8) containing 0.25 $\mu$g of chymotrypsin. Digestion was performed at room temperature with mixing overnight. The chymotrypsin was quenched by adding TFA to a final concentration of 0.5% (v/v). One of the trypsin digests and the trypsin/chymotrypsin digest were desalted using ZipTip $C_{18}$ (Millipore, Billerica, MA, USA) and eluted with 70% (v/v) acetonitrile in 0.1% (v/v) TFA. The desalted material was concentrated to dryness in a centrifugal evaporator.

The proteolytically digested samples were suspended in 20 $\mu$L of 2% acetonitrile (v/v) in 0.1% (v/v) formic acid, and 18 $\mu$L was analyzed by liquid chromatography/electrospray ionization tandem MS (LC/ESI MS/MS) with a Thermo Scientific Easy-nLC II (Thermo Scientific) coupled to an Orbitrap Elite ETD (Thermo Scientific) mass spectrometer using a trap-column configuration as described (*Licklider et al., 2002*). Inline desalting was accomplished using a reversed-phase trap column (100 $\mu$m × 20 mm) packed with Magic $C_{18}$AQ (5 $\mu$m, 200 Å resin; Michrom Bioresources, Auburn, CA) followed by peptide separations on a reversed-phase column (75 $\mu$m × 250 mm) packed with Magic $C_{18}$AQ (5 $\mu$m, 100 Å resin) directly mounted on the electrospray ion source. A 45-min gradient from 2% to 35% (v/v) acetonitrile in 0.1% (v/v) formic acid at a flow rate of 400 nL/min was used for chromatographic separations. A spray voltage of 2,500 V was applied to the electrospray tip, and the Orbitrap Elite instrument was operated in the data-dependent mode, switching automatically between MS survey scans in the Orbitrap (automatic gain control (AGC) target value 1,000,000, resolution 120,000, and injection time 250 ms) with collision induced dissociation (CID) MS/MS spectra acquisition in the linear ion trap (AGC target value of 10,000 and injection time 100 ms), higher-energy collision induced dissociation (HCD) MS/MS spectra acquisition in the Orbitrap (AGC target value of 50,000, 15,000 resolution, and injection time 250 ms), and electron transfer dissociation (ETD) MS/MS spectra acquisition in the Orbitrap (AGC target value of 50,000, 15,000 resolution, and injection time 250 ms). The three most intense precursor ions from the

Fourier transform full scan were each consecutively selected for fragmentation in the linear ion trap by CID with a normalized collision energy of 35%, fragmentation in the HCD cell with normalized collision energy of 35%, and fragmentation by ETD with 100 ms activation time. Selected ions were dynamically excluded for 30 s.

Data analysis was performed using Proteome Discoverer 1.4 (Thermo Scientific). The data were searched against the *Saccharomyces* Genome Database (downloaded 02/03/2011; http://www.yeastgenome.org) that was appended with protein sequences of common contaminants and the sequence for the *Deg1*-Sec62-N153D protein. Searches were conducted as no-enzyme search. The precursor ion tolerance was set to 10 ppm, and the fragment ion tolerance was set to 0.6 Da. Variable modifications included oxidation on methionine (+15.995 Da), carbamidomethyl (+57.021 Da) on cysteine, and acetylation on lysine and any N-terminus (+42.011 Da). All search results were run through the Percolator algorithm for false discovery rate evaluation of the identified peptides.

### Cycloheximide chase, cell lysis, and endoglycosidase H treatment

Cycloheximide chase analysis was performed as described (*Buchanan et al., 2016*). Briefly, mid-logarithmic growth-phase yeast cells were concentrated to 2.5 $OD_{600}$ units/mL in fresh plasmid-selective media, and cycloheximide was added to a final concentration of 250 µg/mL. 950 µL aliquots (~2.4 $OD_{600}$ units) of cells were harvested at the indicated time points following cycloheximide administration and added to 50 µL of ice-cold 20X stop mix (200 mM sodium azide, 5 mg/mL bovine serum albumin) and placed on ice until the end of the chase. Proteins were liberated by the post-alkaline protein extraction method (*Kushnirov, 2000*; *Watts et al., 2015*). To assay protein N-linked glycosylation, protein extracts derived from 0.375 $OD_{600}$ units of cells were supplemented with potassium acetate, pH 5.6 (80 mM final concentration), and incubated at 37 °C for 2 h in the absence or presence of endoglycosidase H (Endo H; Roche Diagnostics, Indianapolis, IN, USA).

### Western blotting

Proteins were separated by SDS-PAGE and transferred to Immobilon polyvinylidene fluoride membranes (Millipore, Billerica, NJ, USA). Membranes were blocked with 5% skim milk in Tris-buffered saline (TBS; 50 mM Tris, pH 7.6, 136 mM NaCl) for 60 min at room temperature or overnight at 4 °C. All antibody incubations were performed for 60 min at room temperature in 1% skim milk in TBS with 0.1% Tween 20 (TBS/T). To detect CPY*-HA, membranes were incubated in the presence of monoclonal mouse anti-HA antibodies (Clone 16B12; Covance, Princeton, NJ, USA) at 1:1,000. To detect Pgk1, membranes were incubated in the presence of monoclonal mouse anti-yeast Pgk1 antibodies (Clone 22C5D8; Life Technologies, Carlsbad, CA, USA) at 1:20,000. Anti-HA and Anti-Pgk1 primary antibody incubations were followed by incubation in the presence of Alexa-Fluor 680-conjugated Rabbit anti-Mouse IgG (H + L) secondary antibodies (Life Technologies, Carlsbad, CA, USA) at 1:40,000. The *Deg1*-Sec62 protein and its derivatives possess two tandem C-terminal *Staphylococcus aureus* Protein A tags (*Hjelm, Sjodahl & Sjoquist, 1975*) and were detected directly with Alexa-Fluor 680-conjugated Rabbit anti-Mouse IgG (H + L) secondary antibodies at 1:40,000. Membranes were imaged using an Odyssey CLx Imaging System (LI-COR, Lincoln, NE, USA).

## RESULTS

### A model unstable translocon-associated protein is acetylated

We performed MS to gain further insight into the nature of *Deg1*-Sec62 PTM. We and others have previously mapped the major N-linked glycosylation site to Asn153 (*Rubenstein et al., 2012*; *Scott & Schekman, 2008*). In our previous work, mutation of Asn153 to a non-glycosylatable aspartic acid residue did not detectably alter translocon engagement, additional PTM, or Hrd1-dependent degradation (*Rubenstein et al., 2012*). Therefore, to simplify PTM analysis, we immunoprecipitated a variant of *Deg1*-Sec62 (*Deg1*-Sec62-N153D) that harbors an asparagine-to-aspartic acid mutation at this position from lysates of cells lacking the Hrd1 ubiquitin ligase. Purified *Deg1*-Sec62-N153D was separated by SDS-PAGE, and bands corresponding to *Deg1*-Sec62-N153D were excised (Fig. 1B) and proteolytically digested. Proteolytic processing of *Deg1*-Sec62-N153D with trypsin was complemented by further digesting one half of the trypsin-digested material with chymotrypsin to reduce large tryptic peptides into smaller sizes conducive to MS analysis. The trypsin and trypsin-chymotrypsin digestions were subjected to MS-based acetylpeptide mapping, resulting in a combined map that covered 73% of the protein sequence (Fig. 1C). Acetylation was identified with high confidence (with a false discovery rate of acetylated peptides of less than 1%) on the N-terminus of the protein, Lys3, and Lys7. Manual visualization of the results verified the high-confidence assignment of acetylation on the N-terminus and Lys3, while manual visualization indicated acetylation of Lys7 should be considered a lower-confidence result, in contradiction to the database search algorithm result. It is possible that additional lysine residues are acetylated but were not detected due to incomplete coverage of the *Deg1*-Sec62-N153D protein. Representative tandem mass spectra identifying sites of acetylation are shown in Figs. 1D–1I.

### Acetylation of N-terminus and two internal residues does not influence Hrd1-dependent degradation of a model unstable translocon-associated protein

To determine if Hrd1-dependent degradation of *Deg1*-Sec62 is influenced by acetylation, we analyzed protein turnover in the context of mutations predicted to prevent acetylation. Certain conditions trigger a switch in E3 dependence for *Deg1*-Sec62 degradation from Hrd1 to Doa10. To evaluate Hrd1-dependent degradation exclusively, we used a version of *Deg1*-Sec62 specifically resistant to Doa10-dependent degradation. This variant, termed *Deg1**-Sec62, harbors two point mutations (F18S and I22T) and behaves like *Deg1*-Sec62 with respect to Hrd1-dependent degradation (*Johnson et al., 1998*; *Rubenstein et al., 2012*). *Deg1*-Sec62 and *Deg1**-Sec62 have been used interchangeably to evaluate degradation requirements for proteins that aberrantly engage the ER translocon (*Rubenstein et al., 2012*).

The N-terminal methionine of *Deg1**-Sec62 is predicted to be acetylated by the NatB N-terminal acetyltransferase, which modifies proteins with the N-terminal Met-Asn dipeptide. Deleting the gene encoding the catalytic subunit of NatB (*NAT3*) is therefore predicted to abolish N-terminal acetylation of *Deg1**-Sec62 (*Hollebeke, Van Damme & Gevaert, 2012*; *Polevoda, Arnesen & Sherman, 2009*). To test the hypothesis that N-terminal acetylation regulates substrate degradation, we analyzed *Deg1**-Sec62 turnover by cycloheximide chase

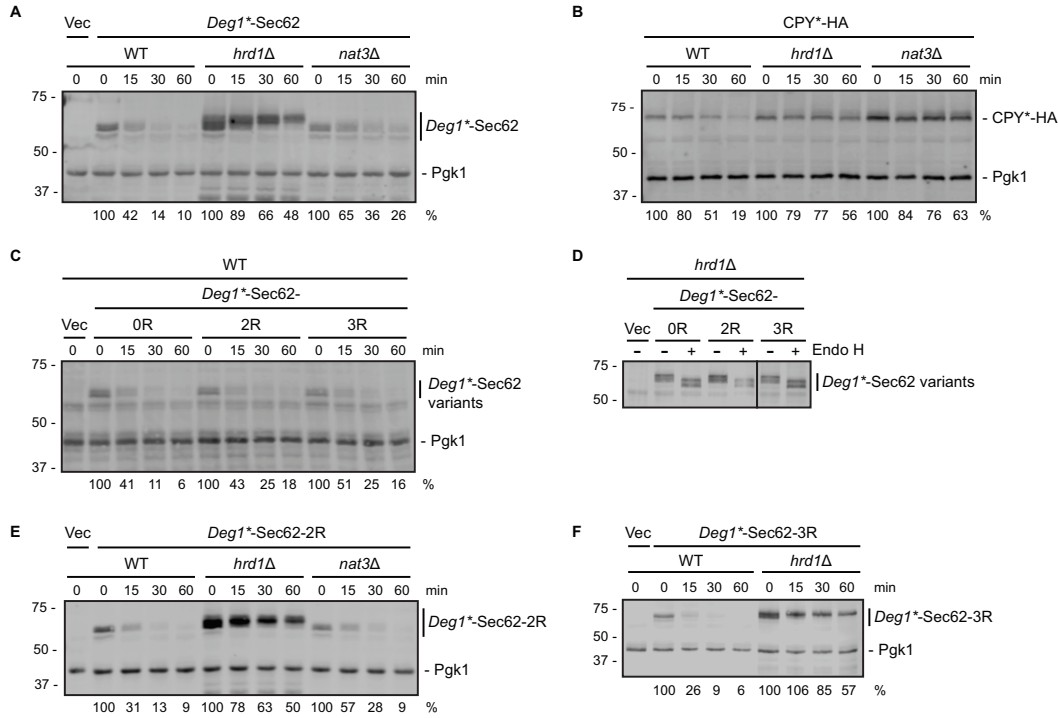

**Figure 2** **Neither N-terminal acetylation nor internal acetylation is required for Hrd1-dependent degradation of *Deg1\**-Sec62.** (A) Cycloheximide chase analysis of yeast cells of the indicated genotypes expressing *Deg1\**-Sec62 or harboring an empty vector. (B) Cycloheximide chase analysis of yeast cells of the indicated genotypes expressing HA-tagged CPY\*. (C) Cycloheximide chase analysis of wild-type yeast cells expressing variants of *Deg1\**-Sec62 or harboring an empty vector. (D) Lysates from *hrd1Δ* cells expressing the indicated variants of *Deg1\**-Sec62 or harboring an empty vector were incubated in the absence or presence of Endo H. The black line indicates that intervening lanes have been omitted. (E) Cycloheximide chase analysis of yeast cells of the indicated genotypes expressing *Deg1\**-Sec62-2R or harboring an empty vector. (F) Cycloheximide chase analysis of yeast cells of the indicated genotypes expressing *Deg1\**-Sec62-3R or harboring an empty vector. Pgk1 serves as a loading control for (A, B C, E, F). For each cycloheximide chase, the percentage of protein remaining at each time point (normalized to Pgk1) is indicated below the image. Experiments depicted in (A, C, E, F) were repeated at least three times, and a representative image is presented for each. The control experiment in (B) (to verify previously published behavior of the tested strains with respect to CPY\* degradation; *Hiller et al., 1996*; *Zattas et al., 2013*) and assessment of Endo H sensitivity in (D) (to confirm protein N-linked glycosylation status) were each performed one time. Vec, Vector. WT, wild-type. 0R, no mutation of acetylation sites. 2R, Lys3 and Lys7 mutated to Arg to prevent internal acetylation. 3R, Asn2, Lys3, and Lys7 mutated to Arg to prevent N-terminal and internal acetylation.

in wild-type yeast, yeast lacking *HRD1*, and yeast lacking *NAT3* (Fig. 2A). Consistent with previous reports, deletion of *HRD1* substantially increased steady state levels of *Deg1\**-Sec62 and delayed protein degradation (*Buchanan et al., 2016*; *Rubenstein et al., 2012*; *Watts et al., 2015*). By contrast, eliminating Nat3 from cells did not markedly impair *Deg1\**-Sec62 degradation. The control ERAD-L substrate CPY\*-HA was stabilized in both *hrd1Δ* and *nat3Δ* cells (Fig. 2B), also consistent with previous reports (*Hiller et al., 1996*; *Zattas et al., 2013*).

We note marginally impaired degradation (without an increase in steady state abundance) of *Deg1\**-Sec62 in *nat3Δ* cells compared to wild-type yeast (Fig. 2A). This

minor effect is likely indirect, as *NAT3* deletion destabilizes the Hrd1 co-factor Der1, loss of which subtly stabilizes *Deg1\**-Sec62 (*Rubenstein et al., 2012*; *Zattas et al., 2013*). Degradation of a *Deg1\**-Sec62 variant with a mutation that abolishes the NatB consensus sequence (*Deg1\**-Sec62-3R in Fig. 2C, described below) supports the notion that N-terminal acetylation does not promote substrate degradation.

To determine if internal acetylation influences *Deg1\**-Sec62 degradation, we mutated both potentially acetylated lysine residues (at positions 3 and 7) to non-acetylatable arginine residues. The resultant protein is referred to as *Deg1\**-Sec62-2R (the version of the protein with both lysine residues intact is called *Deg1\**-Sec62-0R in Figs. 2C and 2D). *Deg1\**-Sec62-2R exhibited degradation kinetics similar to that of *Deg1\**-Sec62 in wild-type cells (Fig. 2C), and its rapid degradation was Hrd1-dependent (Fig. 2E). N-glycosylation of *Deg1\**-Sec62 and its derivatives only occurs following aberrant translocon engagement (*Rubenstein et al., 2012*). Sensitivity to Endo H, which removes all N-linked glycans from yeast proteins, confirmed that *Deg1\**-Sec62-2R engages the translocon in a manner similar to *Deg1\**-Sec62 (Fig. 2D). Therefore, acetylation of two lysine residues is dispensable for Hrd1-dependent degradation of a protein that aberrantly and persistently engages the translocon.

It was formally possible that acetylation at any one of three positions (Met1, Lys3, or Lys7) influences Hrd1-dependent degradation of *Deg1\**-Sec62. We simultaneously abolished all three acetylation events in two different ways. First, we generated a variant of *Deg1\**-Sec62 with point mutations predicted to prevent all three acetylation events. As for *Deg1\**-Sec62-2R, lysine residues at positions 3 and 7 were mutated to arginine. In addition, the second amino acid (asparagine) was mutated to arginine. Proteins with an N-terminal Met-Arg dipeptide are not predicted to be acetylated by any of the characterized N-terminal acetyltransferases (*Polevoda, Arnesen & Sherman, 2009*). The protein with three amino acids mutated to arginine is referred to as *Deg1\**-Sec62-3R. *Deg1\**-Sec62-3R is degraded at a rate similar to that of fully acetylatable *Deg1\**-Sec62 (Fig. 2C) in a Hrd1-dependent manner (Fig. 2F). Endo H sensitivity confirmed that *Deg1\**-Sec62-3R engages the translocon in a manner similar to *Deg1\**-Sec62 and *Deg1\**-Sec62-2R (Fig. 2D). Consistently, degradation of *Deg1\**-Sec62-2R occurred at a similar rate in the presence and absence of Nat3 (Fig. 2E).

## DISCUSSION

Given that PTM of *Deg1*(\*)-Sec62 correlates with its degradation by the Hrd1 pathway, and that acetylation has been shown to influence the degradation of a number of unstable proteins, including ERAD substrates, the hypothesis that acetylation regulates the degradation of a model translocon-associated protein was attractive. However, our data strongly suggest that acetylation of Met1, Lys3, and Lys7 within the model translocon-associated protein *Deg1\**-Sec62 is not required for aberrant translocon engagement or targeting by the Hrd1 ubiquitin ligase.

To purify a quantity of *Deg1*-Sec62-N153D protein sufficient for MS analysis, we took measures to enhance its synthesis and impair its destruction. To increase synthesis, we placed the gene encoding *Deg1*-Sec62-N153D under the control of the galactose-inducible *GAL1* promoter (*Johnston & Davis, 1984*). To reduce degradation, we expressed

the construct in cells lacking *HRD1* (*Rubenstein et al., 2012*). We note that it is conceivable that such alterations alter the manner in which the protein engages the translocon or undergoes PTM. High-level expression of a normally unstable protein might saturate modifying enzymes or be subject to promiscuous modification by PTM machinery.

N-terminal acetylation has been reported to regulate the degradation of multiple ERAD substrates. Deleting the gene encoding the N-terminal acetyltransferase Nat3 has opposing, but related, effects on Hrd1 substrates (*Zattas et al., 2013*). When its Nat3-mediated N-terminal acetylation is prevented, the Hrd1 cofactor Der1 is converted into a Hrd1 substrate. Der1 functions in Hrd1-mediated ERAD-L; thus, proteolysis of Der1 in *nat3*Δ cells stabilizes the ERAD-L substrate CPY*. N-terminal acetylation has also been suggested to affect the degradation of Doa10 substrates (*Hwang, Shemorry & Varshavsky, 2010*; *Ravid, Kreft & Hochstrasser, 2006*). The MATα2 transcriptional repressor, from which the *Deg1* degron is derived, is N-terminally acetylated in a Nat3-dependent manner (*Hwang, Shemorry & Varshavsky, 2010*). Reports suggest that Doa10 targets MATα2 and other ERAD-C substrates for degradation in a manner that requires N-terminal acetylation as a part of the N-end rule system. However, the extent to which acetylation affects Doa10 substrate degradation varies among reports (*Hwang, Shemorry & Varshavsky, 2010*; *Ravid, Kreft & Hochstrasser, 2006*; *Zattas et al., 2013*).

Internal acetylation has also emerged as a positive and negative regulator of protein degradation (*Gronroos et al., 2002*; *Nguyen et al., 2016*). Acetyltransferases modifying internal lysine residues exist in the nucleus, cytosol, and ER (*Pehar & Puglielli, 2013*). In cases where internal modification enhances degradation, acetylation may stabilize the interaction between substrate and ubiquitin ligase (*Jiang et al., 2011*; *Nguyen et al., 2016*). In cases where internal modification reduces degradation, acetylation may compete with ubiquitylation for lysine modification or prevent substrate binding to ubiquitin ligases (*Caron, Boyault & Khochbin, 2005*; *Chang et al., 2016*). Interestingly, gp78-dependent binding to and ubiquitylation of the ER chaperone GRP78 is enhanced by substrate deacetylation (*Chang et al., 2016*). Given the abundance of acetylated proteins at the ER (*Pehar et al., 2012*), it would not be surprising if acetylation regulates ERAD of a subset of unstable proteins. Further, acetylation of apoB, a physiological protein that exhibits prolonged translocon association, influences uptake by target cells (*Brown et al., 1980*; *Brown et al., 1979*; *Weisgraber, Innerarity & Mahley, 1978*). To our knowledge, the role of acetylation in apoB degradation has not been explored.

The molecular mechanism by which Hrd1 recognizes the model translocon-associated protein *Deg1*-Sec62 remains obscure. Several protein co-factors that function with Hrd1 in the destruction of other classes of ERAD substrates are dispensable for *Deg1*-Sec62 degradation (*Rubenstein et al., 2012*). Additionally, while post-translational modification of this protein correlates with its Hrd1-dependent destruction, no modifications (save ubiquitylation) have been shown to influence its degradation (*Rubenstein et al., 2012*; *Scott & Schekman, 2008*). Ongoing investigations are being conducted to identify the factors involved in recognition and degradation of proteins that persistently and aberrantly engage the ER translocon. Such factors may represent therapeutic targets for cholesterol-related pathology.

## CONCLUSIONS

We have shown that a model unstable translocon-associated protein is N-terminally and internally acetylated. Despite reported roles for acetylation in the regulation of degradation of other unstable proteins, including multiple substrates of ER-resident ubiquitin ligases (*Gronroos et al., 2002*; *Hwang, Shemorry & Varshavsky, 2010*; *Nguyen et al., 2016*; *Ravid, Kreft & Hochstrasser, 2006*; *Zattas et al., 2013*), acetylation does not influence the degradation of the protein investigated in this study. Our results demonstrate that context is crucially important in determining whether a particular PTM will trigger degradation by the UPS. Indeed, while N-terminal acetylation status may seal the degradative fate of several proteins (e.g., via the N-end rule pathway; (*Hwang, Shemorry & Varshavsky, 2010*), our data indicate that such rules are not absolute and emphasize that even the most reasonable predictions must be empirically tested.

## ACKNOWLEDGEMENTS

We thank Mark Hochstrasser (Yale University; New Haven, CT) for sharing yeast strains and plasmids. We thank Christopher Hickey (Yale University; New Haven, CT) and Robert Tomko (Florida State University; Tallahassee, FL) for invaluable technical guidance. We thank Philip Gafken (Fred Hutchinson Cancer Research Center; Seattle, WA) for providing outstanding experimental assistance and support for this project. We thank Bryce Buchanan for assistance in data analysis. We thank present and former members of the Rubenstein laboratory for providing an enthusiastic and supportive environment.

### Funding

This work was supported by a National Institutes of Health (NIH) Grant R15 GM111713, a Ball State University ASPiRE Junior Faculty Research Award, and funds from the Ball State University Provost's Office and Department of Biology to EMR. There was no additional external funding received for this study. The funders had no role in study design, data collection and analysis, decision to publish, or preparation of the manuscript.

### Grant Disclosures

The following grant information was disclosed by the authors:
National Institutes of Health (NIH): R15 GM111713.
Ball State University ASPiRE Junior Faculty Research Award.
Ball State University Provost's Office and Department of Biology.

### Competing Interests

Sarah M. Engle is an employee of Eli Lilly and Company (Indianapolis, Indiana, USA), and Samuel Z. Coffey is an employee of Medpace Reference Laboratories (Cincinnati, Ohio, USA).

## Author Contributions

- Sarah M. Engle, Justin J. Crowder and Sheldon G. Watts performed the experiments, analyzed the data, contributed reagents/materials/analysis tools, reviewed drafts of the paper.
- Christopher J. Indovina performed the experiments, analyzed the data, reviewed drafts of the paper.
- Samuel Z. Coffey performed the experiments, contributed reagents/materials/analysis tools, reviewed drafts of the paper.
- Eric M. Rubenstein conceived and designed the experiments, analyzed the data, wrote the paper, prepared figures and/or tables, reviewed drafts of the paper.

## Data Availability

Engle, Sarah; Crowder, Justin; Watts, Sheldon; Indovina, Christopher; Coffey, Samuel; Rubenstein, Eric (2017): Engle_BLANK.raw. figshare.

10.6084/m9.figshare.5119870.v1.

Contains mass spectrometry files (.raw) and Proteome Discoverer protein database search results (.msf).

## Supplemental Information

Supplemental information for this article can be found online at http://dx.doi.org/10.7717/peerj.3728#supplemental-information.

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
