# Peer review of "Acetylation of N-terminus and two internal amino acids is dispensable for degradation of a protein that aberrantly engages the endoplasmic reticulum translocon"

_PeerJ, doi:10.7717/peerj.3728_

## Round 0.1 · original submission · Minor Revisions

As you can see from the reviewers' comments, all reviewers agree that the manuscripts is clearly written and the experiments carefully conducted. They suggest some textual adjustments and clarifications to improve presentation and context for readers. These, together with a quantification of the chase experiments as suggested by reviewer 2, seem reasonable requests that are straightforward to execute. I look forward to receiving a revised manuscript, addressing these minor issues.

Reviewer 1 ·

Basic reporting

This manuscript is well written. The appropriate background is presented and the data interpreted in a concise and clear manner. The structure of the article, the figures, and tables is professional.

In the abstract (line 25-28) and introduction (line 72-75) the description of ApoB degradation is too general and needs to be clarified, mainly regarding the degradation in yeast versus mammalian cells. As written it sounds like ApoB is degraded by Hrd1 in both yeast and mammalian cells. Although ApoB is degraded by Hrd1 in yeast, it is degraded by gp78 in mammalian cells. This is important to clarify to avoid confusion.

Experimental design

The research question is well defined, rigorously tested, and sufficient detail provided.

Validity of the findings

Data is robust and conclusions are well stated. Although abrogation of the acetylation state in the nat3∆ mutant is directly confirmed by mass spec, this seems likely and is not considered a major weakness. Combined with the mutation analysis the conclusion that Deg1*-Sec62 does not require acetylation is well supported.

Additional comments

This is a very nice paper. Well constructed and clearly written. It is also to important to the field to point out that acetylation is not always a major determinant of degradation. Future studies to tease out these details around acetylation and protein stability will be of interest.

·

Basic reporting

No major comments. Manuscript is well written and structured clearly throughout. I would appreciate it though if the authors included a short paragraph on protein acetylation in the introduction or discussion. In particular, what is known about the machinery that mediates internal K-acetylations? Is there any link between this machinery and ERAD, maybe genetic interactions? Furthermore, is there any indication in the literature that acetylation of the substrate plays a role in Hrd1-mediated ERAD? Please comment.
Minor points:
line 37: “fate” cannot be used as a verb, I think
line 43: akronym UPS, write full name once
line 361: post-translationally should be post-translational, I think

Experimental design

No comments.

Validity of the findings

There are, in my opinion a few caveats that should be mentioned in the manuscript:
A hrd1 deletion strain is used for purification of the substrate. Expression is carried out under Gal1 promoter control.
Usage of the deletion strain is understandable but carries the risk that the substrate isolated is different from the one that is the target of investigation in this study, namely a translocon-stalled version of Deg1-Sec62. In principle these could be differentially modified, if modification is changed in response to Hrd1-engagement. Please comment.
Expression of the substrate under Gal1 control is, again, totally understandable. However this leads to a massive overexpression. Is there a possibility that this overexpression could alter post-translational modifications, like acetylation? Is saturation of acetylation machineries a concern at all? Please comment.
Furthermore, in my opinion, analysis by mass-spectrometry is not conclusive on the question of how much of the substrate carries a certain modification. In other words, can the authors estimate how much of the substrate is acetylated at certain sites? I am not a mass-spectrometry expert, hence, this suggestion might be useless: Can you compare abundance of the modified and non-modified peptides? The results cannot rule out that other internal sites, not covered by mass-spec analysis become acetylated. While the authors discuss the results very carefully in this respect, this caveat should be explicitly stated, in my opinion.
In line 297, the authors state that degradation of the substrate was marginally impaired in the nat3 deletion strain. Could you please provide a quantification of the chase experiments, possibly with averaged results.

Additional comments

Thanks very much for taking the effort to share these results with the scientific community.

Reviewer 3 ·

Basic reporting

The manuscript is well written and provides sufficient background information for general readers. The figures and tables are well displayed, and the results are relevant to the hypothesis.

Experimental design

The research question was well defined. The techniques applied in this study meet a high standard. The methods were detailed.

Validity of the findings

Data is sound and in line with the author's conclusion.

Additional comments

The acetylation sites (M1, K3 and K7) were identified in a Deg1-sec62-N153D mutant, which lacks glycosylation. However, the degradation assays were conducted in a wildtype version (either Deg1-Sec62 or Deg1*-Sec62). It is unclear whether M1, K3 and K7 are acetylated at all in the wildtype version. If not, this study would appear to address a non-existing question. The author may want to clarify this issue.

---

## Round 0.2 · accepted · Accept

Thanks for sending the revised manuscript and addressing all concerns.

Reviewer 1 ·

Basic reporting

No comment.

Experimental design

No comment.

Validity of the findings

No comment.

Additional comments

All comments were satisfactorily addressed and this manuscript is now ready for publication.

·

Basic reporting

All my concerns have been addressed appropriately.

Experimental design

All my concerns have been addressed appropriately.

Validity of the findings

All my concerns have been addressed appropriately.